# Design of an SMA-Based Actuator for Replicating Normal Gait Patterns in Pediatric Patients with Cerebral Palsy

**DOI:** 10.3390/biomimetics9070376

**Published:** 2024-06-21

**Authors:** Paloma Mansilla Navarro, Dorin Copaci, Janeth Arias, Dolores Blanco Rojas

**Affiliations:** Department of Systems Engineering and Automation, Universidad Carlos III de Madrid, 28911 Leganes, Spain; dcopaci@ing.uc3m.es (D.C.); jaariasg@pa.uc3m.es (J.A.); dblanco@ing.uc3m.es (D.B.R.)

**Keywords:** biomedical engineering, biomimetic actuators, neuro-rehabilitation, wearable exosuits, shape memory alloys

## Abstract

Cerebral Palsy refers to a group of incurable motor disorders affecting 0.22% of the global population. Symptoms are managed by physiotherapists, often using rehabilitation robotics. Exoskeletons, offering advantages over conventional therapies, are evolving to be more wearable and biomimetic, requiring new flexible actuators that mimic human tissue. The main objective behind this article is the design of a flexible exosuit based on shape-memory-alloy-based artificial muscles for pediatric patients that replicate the walking cycle pattern in the ankle joint. Thus, four shape-memory-alloy-based actuators were sewn to an exosuit at the desired actuation points and controlled by a two-level controller. The loop is closed through six inertial sensors that estimate the real angular position of both ankles. Different frequencies of actuation have been tested, along with the response of the actuators to different walking cycle patterns. These tests have been performed over long periods of time, comparing the reference created by a reference generator based on pediatric walking patterns and the response measured by the inertial sensors. The results provide important measurements concerning errors, working frequencies and cooling times, proving that this technology could be used in this and similar applications and highlighting its limitations.

## 1. Introduction

Cerebral Palsy (CP) is defined as a group of syndromes or motor disorders with a common origin in neural activity. Its comorbidities often include epilepsy, learning difficulties, conductive anomalies and sensory disorders. It impacts approximately 0.2177% of the worldwide population and has no possible cure [1]. CP symptoms can be treated through physiotherapy, speech therapy, educational support and occupational therapy, enhancing the individual’s functional capabilities and life quality. The treatment depends on the dysfunction pattern, with higher impacts from physical and occupational therapy [2].

During the Discover2Walk (D2W) Spanish national project, this research group within the RoboticsLab (Universidad Carlos III de Madrid, UC3M), along with the ’Consejo Superior de Investigaciones Científicas (CSIC)’, aimed to develop a flexible exosuit along with a DC-motor-based walker to provide a walking platform for children affected with CP [3]. The main objective was to provide physical support to these children during the early stages of the walking cycle learning process (focused on weight-suspended passive rehabilitation). For this purpose, pediatric patients were assisted at an age similar to that when they would have started walking if they did not suffer from CP. The target subject was a 3-year-old patient (there are no exoskeletons on the market for such a young age yet) with a GMFCS (Gross Motor Function Classification System) score equal or lower than level III [4,5]. These types of patients are able to walk in limited environments when they are old enough, but almost never in an autonomous way. They could greatly benefit from the continuous repetition of walking patterns in each of the joints of the lower limb due to the high neuroplasticity present at early ages [6,7].

Currently, manual interventions do not solve the walking limitations experienced by these patients efficiently; exoskeletons have been proven to be accurate for helping pediatric patients with CP in the walking cycle through smooth and accurate movements [8]. Some exoskeletons have been developed and even commercialized with this purpose, involving patients older than 6 years old [9,10]. However, nowadays, these exoskeletons are limited to research environments due to their complexity, price, weight and size [11,12,13]. Moreover, the misalignment between the user and the exoskeleton joints can negatively affect patient mobility [12], creating unintentional interference with the natural biomechanics of the body [14]. Pons [15] states the importance of working with more biomimetic and less rigid architectures, trying to mimic the human biomechanics. These new devices are known as soft exoskeletons or exosuits [16]. Compared to rigid exoskeletons, exosuits present certain challenges related to the actuation force, specifically the transfer of force to the desired points, the actuator weight and the complexity of the design (the human body is compliant and cannot support high pressures [14]). To the best of the authors knowledge, exosuits have been developed for the adult population exclusively. For example, Zhao et al. [17] present the design, modeling and control of an ankle exosuit that aids in the plantar–flexion movement during gait locomotion. The exosuit design is based on a muscle–tendon–ligament model and is actuated by two DC motors, where the weight of the motors and gear was 1.5 kg. In [18], an exosuit for plantar–flexion based on cable-driven actuators with admittance control was developed and successfully tested with six post-stroke subjects; the total weight of the system was 4.93 kg. All of the patients required an actuation box, normally placed on the patients back, to support this extra weight, reducing the patients’ comfort.

Hence, brand new, lighter actuators are necessary, especially when applied to the pediatric population, whose weight and force are lower compared with the adult population. These actuators should mimic human biomechanics and performance, turning into biomimetic soft actuators. To meet the low price, weight and size requirements, shape memory alloys (SMAs) [19] were selected. SMAs are made out of nickel and titanium and change their inner structure when subjected to a certain temperature, generating a longitudinal contraction and, thus, a displacement in the actuator trajectory. Furthermore, this material has proven to work well in exoskeleton actuation [20,21,22], with potential applications in exosuits. On the other hand, new sensors that do not depend on fixed axes of motion, like absolute rotation sensors, are needed to evaluate the angular position of the ankle joint. Various tracking systems, such as optical methods (e.g., BTS smart, Vicon), use markers at body points to digitize limb movements. These systems, however, require a lab environment. Advances in electronic and mechatronic technologies have led to the development of inertial measurement units (IMUs) which include accelerometers, gyroscopes and magnetometers. These portable and robust devices can measure and record human gait parameters [23].

This article focuses on an experiment carried out in the RoboticsLab, who were in charge of developing SMA-based actuators along with the exosuit design. As part of the D2W project, both the knee and ankle joints were actuated through an SMA, with the patient’s weight completely suspended. The knee design is covered in [13]. The ankle design, covered in this article, was based on previous developments of these actuators, completely reconfigured in terms of length, routing and control parameters for this project, evidencing the potential use of SMAs as artificial muscles or soft actuators. Moreover, a whole new software was developed for both the sensors and the SMA control. This device worked together with a DC-motor-based end-effector linked to the leg immediately above the ankle joint developed by the CSIC. The Bowden actuation system developed along with the textile components is weightless and does not need additional devices to support the extra load. Furthermore, a dummy was designed in order to test the different prototypes developed, as well as the final device. This dummy represented a 3-year-old patient, with mobility in the sagittal plane in both the knee and ankle.

The article is divided into five sections. Section 2 resumes the design of the dummy along with the exosuit developed by the UC3M, focusing on the soft actuator architecture and control. Section 3 gathers the software development behind its actuation. Section 4 presents the main control tests of the actuator integrated in the exosuit, with promising results that prove its possible use in slow rehabilitation therapies. Finally, Section 5 discusses the main conclusions and future guidelines.

## 2. Materials and Methods

### 2.1. Metrics

In order to calculate the parameters needed to actuate different patients’ joints, it was essential to gather data from the literature concerning the length and weight measurements of a 3-year-old patient. Data were obtained from healthy individuals due to the lack of this type of information in patients suffering from CP at early stages. From these data, the torques generated in each joint were estimated, along with the percentage of shortening needed by the SMA wire to provide the desired actuation. Table 1 shows the measurements obtained from the literature [24,25,26,27].

Furthermore, functional angles during the walking cycle were obtained for both the knee and the ankle (Table 2). This information was used, both for dummy construction and parameter calculation [28,29].

For the dummy construction, the main bones conforming the leg were modeled in Standard Triangle Language (STL). The femur, tibia and fibula were obtained from medical repositories. The STL of the foot bones was designed from a DICOM file from a Computerized Axial Tomography (CAT) scan of an anonymous patient through different vision techniques. STLs were resized to fit the values in Table 1 and printed in polylactic acid (PLA) (Figure 1).

These PLA prototypes were covered in platinum silicone (PlatSil Gel-25) with a shore of A25 [30]. Its shore and density (1.107 g/cm^3^) are very similar to those in human muscles (1.0597 g/cm^3^) [31], which constitute the main volume and weight of soft tissues [32]. In order to model this silicone, it is necessary to use casts built similar to bones, shown in Figure 1 and printed in PLA. The measurements of the dummy are also covered in Table 1; errors are minimal when compared with real parameters.

Due to the project specifications, only one movement was controlled during the device actuation: ankle plantar and dorsal flexion. Additionally, the knee flexo-extension needed to be evaluated to avoid possible undesired forces created by the ankle actuator. Hence, the dummy had four degrees of freedom (two in each leg) restricted to the sagittal plane (maximum angles defined in Table 2). For the knee joint, two hinges were designed; each one linked together the external face of the femoral condyle and the external face of each tibial plateau. The rotation axis was fixed to the femoral epicondyle according to [33]. Furthermore, the attachment point from the hinge to the tibia was created at the same height where the cruciate ligament is inserted, with the joint axis aligned to the insertion of this ligament into the femur [34]. In the ankle joint, a unique piece in the shape of a hinge was used. The rotation axis was fixed to the point where the tibia and the fibula are connected to the talus bone, as these three bones form a natural hinge at this point with the same axis of rotation [34] (see Figure 2). Finally, the result was linked to a dummy with the same dimensions of a 3-year-old child. The specifications concerning the device design highlighted that the objective provides a therapy where the full body weight is suspended by a harness. Because of this, the dummy’s body weight was not relevant in any test or validation phase.

### 2.2. Actuator Computation

Data concerning the center of mass and the inertial moments of the dummy were obtained from [35] and were used to compute the actuation force torques needed in each of the joints. Force and torque requirements were overestimated to make sure they would not break down if an external force affected the performance.

As mentioned before, actuators should mimic the natural structures present in the human body. For this purpose, an exhaustive analysis of biomimetic actuators was performed. As mentioned by Copaci et al. [36,37], SMAs can work as artificial muscles, contracting and elongating their initial shape. These materials have the capacity to recover their initial shape after having been deformed if they are subjected to a specific physical process (in this case, they were subjected to a certain activation temperature (AT)). The alloy selected is made out of nickel and titanium [38], which can be shortened up to 4% of its original length when heated higher than its AT (70 °C or 90 °C). The SMA settings for this specific application are based on the metrics obtained by [35] and the material datasheet [38] covered in Table 3.

Each actuator assembly was based on [37] and their schemes are covered in Figure 3. They consist of an SMA wire covered by a Teflon tube that isolates it thermally and electrically and by a Bowden cable that aims to convert the wire shortening into an actuation movement. The wires were sewed to the exosuit at one end and fixed at the other thanks to the metallic piece shown in Figure 3 and designed within this research group. Both fixations enabled the actuator to perform as an artificial muscle that, when contracted, pulls the body segment attached to it via an artificial tendon, creating a flexion or extension movement in the joint. Two actuators were used for each joint (one worked in flexion and the other in extension), creating an antagonistic configuration in each of them. Hence, four SMA actuators were used (two for each ankle).

Computation concerning the optimal actuators lengths and attachment points to the exosuit was based on [35] and optimized by the authors of this article based on the experimental results obtained through validation tests. This optimization was achieved based on the fulfillment of the specified movement ranges in Table 2.

Finally, the wires were heated based on the Joule effect. This effect demonstrates that, if an electrical current is running along a conductor, part of its kinetic energy is turned into heat. The specifications relative to the intensity needed by each of the wires and the resistance depending on their length are covered in the datasheet of the material and in Table 3. The calculus concerning the voltage needed by each of the actuators to work properly was based on (Equation 1).
(1)$$V=I\cdot R_{SMA}=I\cdot R_{l}\cdot l_{SMA}$$
where *V* represents the voltage needed; *I* is the nominal intensity of the actuator; $R_{SMA}$ is the resistance of the actuator; $R_{l}$ is the nominal resistance of the actuator depending on its length; and $l_{SMA}$ is each wire’s length. Ankle actuators need:(2)$$V=I\cdot R_{l}\cdot l_{SMA}=2.25\cdot 8.3\cdot 1.5=28.0125\;V$$

### 2.3. Exosuit Design

Actuators were stitched to the suit as shown in Figure 4, mimicking the bone position in relation to the human muscles in charge of the specific movements to be accomplished. The exosuit consists of a neoprene suit that isolates the user thermally and electrically. It can be adapted to different bodies and sizes due to its flexibility. Moreover, some zips and clamps were included to enable the adaptation of the suit to different physiognomies.

The literature states the importance of walking barefoot during the initial years of the human development, favoring muscle development in the foot and ankle structures and enhancing foot arch formation. Furthermore, having the foot sole in contact with different and irregular surfaces is beneficial for stimulating kinetic sensations as well as natural reflexes; it is also beneficial for developing proprioception and for improving joint alignment [39].

In order to combine the ankle movements, a device that covers the whole foot and enables its pulling during dorsal and plantar flexion was necessary. Hence, walking barefoot was unfeasible, requiring an assisting shoe. In order to mimic this barefoot walking cycle, a flexible material (a neoprene sock that isolated the foot due to the great temperatures reached by the SMA) was used, in contrast to some rigid shoes used in conventional exoskeletons. This sock allowed for complete mobility of the feet inside the shoe. Naturally, the feet physiognomy of each exo user must be considered in order to adapt the design to possible problems that patients with CP often present [2].

### 2.4. Hardware

*Power Unit.* As shown in Section 2.2, in order to obtain the desired temperature in the SMA wires, they were fed by a voltage difference defined by Equation (Equation 2). The controller that governed the behavior of the wires is described in Section 3.2. It generates a PWM signal that, when attached to the power unit and powered by a power supply, regulates the intensity passing through each of the SMA wires.

*Inertial Sensors (IMUs).* In order to control the flexion and extension generated in each of the joints, it was crucial to measure the flexion angles generated in the ankle of the user (future developments will include the knee joint; hence, the knee angles will be computed as well). For this purpose, three inertial sensors were used in each leg and directly sewed to the exosuit, as can be observed in Figure 4. One was attached to the foot segment, another was attached to the tibial segment and the last one was attached to the thigh segment. BNO-055 IMUs were used, and their datasheet is presented in [40]. The sampling frequency of this sensor within the program is 5 × 10^−3^ s, its range ±250° s^−1^, and its resolution is 16 bits, so its minimum resolution is 7.6 × 10^−3^° s^−1^ per step.

*Discovery STM-32F407.* In order to incorporate all the elements of the hardware, a Discovery STM-32F407 was used. This board contains a STM32F407VGT6 microcontroller which was programmed through the COM ports in it. The scheme of the connections from the target to the rest of the hardware elements is covered in Figure 5.

## 3. Software

The software governing the device’s behavior combines different programming languages and architectures. The aforementioned microcontroller is programmed in C language and was in charge of reading the BNO-055 inertial sensors, translating this information into angular measurements for the ankle joint and providing control signals to the power unit. Communication with the STM was achieved through the serial ports within an NVIDIA Jestson Nano node programmed in C++20 (see Figure 5). This node communicated with other nodes in charge of generating the reference patterns, writing and plotting the information received through ROS2 (https://www.ros.org/, accessed on 5 May 2024). Hence, the tests in this manuscript were performed by the Jetson Nano, but they could be conducted on any computer with access to ROS2 and to a local network. Finally, the main elements involved in device performance can be split into two categories: one for elements in charge of the inertial sensors and one for elements in charge of actuator control.

### 3.1. Inertial Sensors

Communication with the BNO-055 inertial sensors is achieved through an I2C protocol. Six IMU sensors need to be linked, so three I2C modules are needed (each one could receive information from two sensors).

Information coming from these sensors is pre-processed and filtered (low-pass filter). In between the information handover, quaternions are selected because they generate fewer artifacts than the Euler angles. They are turned into flight angles (Yaw, Pitch and Roll). Afterwards, the Pitch angle is selected to isolate the movement difference in the sagittal plane. The Pitch angle was selected due to the way in which the sensors were placed in the suit. Finally, in order to obtain the rotation angle in each joint, the difference between the Pitch angles of the segments adjacent to this joint is computed. Sensors are calibrated statically, taking the upright position of the user as the starting point. Moreover, the results concerning the performance of the IMUs in this device were proven to be feasible in [41].

### 3.2. Control

The control of the device is based on a two-level controller. Additionally, a user interface directly governed by the user indicates which actuators are to be controlled and which is the desired working mode. A high-level controller translates these working modes into reference signals for each of the actuators. Finally, different low-level controllers generate the control signals governing each of the SMA wires’ behavior.

#### 3.2.1. User Interface

The user interface is directly governed by the user and enables the selection of the working mode followed by the exo during the rehabilitation process. For each of the working modes, it can be specified that each actuator works in an only-flexion, only-extension or antagonist configuration. Through the interface, the user can also select when to calibrate the sensors, when to cut the actuators power or when to switch between different working configurations (see Figure 6).

These working modes can involve every joint, synchronizing the movement of both ankles at the same time, each following its own pattern. The basic working mode involves following the standard walking cycle of a 3-year-old child at different velocities (from 1 km/h to 5 km/h), based on [42], which states that every individual’s walking cycle is different depending on their height and velocity. The signal shape is replicated, but its period is manually selected by the user. 

#### 3.2.2. High-Level Controller

The high-level controller is in charge of translating these working modes into reference signals that are sent to each of the actuators. Moreover, the pattern generated for each joint by the middle controller is synchronized with the rest of the joint’s patterns. When any of the actuators are off, the signal generator sends a null signal to its lower-level controller, generating a null control signal. During the calibration periods, the actuators are automatically powered off.

The signals generated for the walking patterns (from 1 to 5 km/h) are based on the child’s height and the calculations performed in [42]. Figure 7 shows the different reference signals for the ankle joint during the different phases of the walking cycle. The horizontal axis represents the percentage of this cycle, translated into the signal period [*T*] depending on the user’s needs. The opposite ankle follows a similar pattern but phased $\frac{T}{2}s$. The ankle signs are positive for dorsiflexion (flexion) and negative for plantar flexion (extension).

#### 3.2.3. Low-Level Controller

Finally, these references are handed to a lower-level controller which, by comparing them to the real angle of each joint (provided by the IMU sensors), generates the angular position error provided to the B-PD regulator.

Each of the low-level controllers governs the actuation of one SMA wire and its antagonist (both of them are never powered simultaneously because they perform opposite actions). The control scheme is based on an equation defined in [43,44] and it is covered in Figure 8. This scheme consists of two bilinear PD controllers (B-PD); each of them generates the control signal for one SMA wire or its antagonist. Due to the fact that the SMA has hysteretic behavior at its activation temperature [19], a bilinear term was incorporated in order to compensate for this non-linear behavior, linearizing its actuation. The parameters of each B-PID controller were adjusted empirically and are different for each of the adjusted controllers.

$U_{ref}$ represents the reference created by the middle controller, which is transformed into $U_{ref\;flexion}$ and $U_{ref\;extension}$ for each controller; the nomenclature for flexion and extension represents each control loop for each actuator. These references are compared with the position measured by the IMUs and fed back into the PD regulator and the bilinear compensator below. Both $V_{f/e}$ and $U_{f/e}$ represent both the control action after the regulator and after the bilinear compensator, respectively. Finally, $Y_{f}$ and $Y_{e}$ represent the wire position after actuation and $Y_{out}$ represents the joint angle measured by the IMUs.

## 4. Results

Results were obtained from tests with the following structure. They are based on the performance of the SMA wires themselves as well as the performance of whole structure, considering the attachment points and the selected parameters.

Duration: 10 min. Tests were designed based on the relevant application. As long as the main purpose of this device is to provide a smooth and continuous performance, tests should last for a certain amount of time. The specific selection of 10 min is based on previous tests performed in the research group. In order to be able to compare these results with other devices and tests, the time must be standardized.The error considered was the mean error in absolute terms of the whole cycle, eliminating the first 30 s which were considered variable due to actuator stabilization. This stabilization was set up to 30 s because this was the longest signal period analyzed and it was necessary to ensure that at least one whole period had finished before starting to measure the error.The PWM signals generated were also gathered due to the important information provided by them.Many patterns were analyzed in these tests. The main idea was to provide a wide range of parameters concerning the frequency of actuation along with the error achieved for each pattern proposed (1–5 km/h patterns). It was crucial to generate a sufficient amount of information that can be used by professionals to select working modes.

### 4.1. Following Each Velocity Pattern at Different Periods

The results in terms of mean errors are covered in Table 4. As previously mentioned, these results were calculated in 10 min tests. However, 30 min tests were performed randomly over different periods and patterns to make sure that the results can be transposed to longer periods of time (the errors were still calculated in 10 min tests to ensure the normalization of these results). Table 4 shows the period in which the actuator overheated and broke down in bold. The maximum frequency (lowest period) at which the actuators can perform without over-heating and breaking is the one immediately after. This value was obtained from experimental results in the laboratory. The lowest periods achieved were also tested over 30 min to ensure that edge results could be transposed to longer periods of time.

The results are visually explained in Figure 9, where it is easier to observe the differences between the different periods. It is obvious that the higher the velocity of the pattern, the higher the errors and the lower the periods at which the actuator can work. This is due to the fact that, at higher velocities, the patterns reached bigger maximums and minimums and changes in time became steeper. Thus, errors became bigger due to the lag between the signal and the response, and, as a consequence of both, PWM signals became higher and overheating occurred sooner. It can be observed that even though the frequency achieved is accurate for the current application, further applications in more advanced states of therapy may need higher actuation frequencies, limited by the actual technology.

Some examples are shown in the figures below (Figure 10 and Figure 11) concerning the behavior of the actuator when following the patterns; the explanations concerning the error shown in Table 4 and the PWM signals prove that the control signal was smooth. Higher values of the error and PWM signals appeared in the 5 km/h pattern tests, so the results are illustrated with a T=15 s and T=12 s 5 km/h pattern tracking. A T=15 s 1 km/h signal is also shown to analyze the differences. Even though the results covered 10 min, the figures cover between 200 s and 300 s of the results to simplify the visualization.

As mentioned before, the higher the velocity of the pattern, the bigger the maximums and minimums reached by the patterns and the steeper the changes in time. Thus, errors became bigger due to the lag between the signals. This can be found in Figure 12, where zooming into the image shows how the errors are mainly due to the phase lag between signals (the peak value differences are lower than 1° and, yet, the errors come up to 6°). This phase lag increases as the reference signal period is reduced due to the faster and steeper changes in its shape. As a result, all these images share a common delay. This delay is mostly due to the elastic behavior of the exosuit, and, when measured, results in a mean value of 0.5 s and a variability of 0.03 s—this is what makes it very similar in every structure.

Finally, the PWMs of both signals are analyzed in Figure 13. It can be observed that the PWMs activating and deactivating each actuator did not intersperse in time at high frequencies, leading to smooth control of the actuator (actuators were not activated and deactivated continuously, leaving enough time for the wire to cool down). Additionally, as expected, the PWM signals were higher for the 5 km/h pattern because the error was greater. Thus, overheating occurred earlier and the actuators could not achieve the same working frequencies as when tracking the 1 km/h pattern.

Figure 14 represents the tracking of the 5 km/h pattern just before overheating. This overheating is observed because the antagonist actuator could not reach the actuation peaks and the maximums started decreasing in subsequent cycles.

### 4.2. Following a Sinusoidal Pattern

To broaden the possibilities and applications of this device, tests were performed with sinusoidal signals as references. The maximum, minimum and period of each reference can be manually selected by the user and were incorporated into software selection.

The results covered in this section are based on a sinusoidal signal with the same maximums and minimums as the 3 km/h pattern, but tests can be performed with any of them. The main idea behind this section is to prove that while working with sinusoidal references, frequencies can be widely enhanced with respect to the pattern.

As can be observed in Table 5, the period of the signal that the actuator can follow when working with sinusoidal references is widely enhanced, as well as the errors within the cycle. Tests were performed over 10 min.

This approach offers new possibilities, e.g., moving the patient’s ankle in the same ranges of movement that were achieved with the walking patterns but in smoother transitions. Hence, mobility and strength could be trained before facing steeper transitions of patterns and their complexity. However, other aspects such as the suspended body weight should be considered for future applications of the device to enhance future states of therapy.

As shown in Figure 15, the error came mainly from the phase lag once again. The error between peaks was lower than 0.5° during the whole test and the PWM signal stayed within low limits (generally under 60 %) and without mixing both the flexion and extension actuation in time. The results are shown between 200 and 300 seconds to ease understanding. Finally, the breakage pattern of the SMA actuator is shown in Figure 16.

### 4.3. Working Improvements

Two different approaches were taken in order to enhance the working frequency achieved. The main idea was to analyze the reasons why the actuator overheats and try to eliminate this overheating.

The first approach is based on the PWM analysis. It can be observed that when PWM values reach greater peaks, overheating occurs. Thus, the cause of overheating could be due to these PWM peaks, or these PWM peaks could be a consequence of overheating. It was interesting to analyze the possibility of limiting the PWM values to 50% (other percentages have been studied but this percentage led to clearer results for data analysis) and check if overheating occurred before, after, or did not occur at all. The approach takes a sinusoidal wave at its breakage period (T = 5 s), shown in Figure 16. Its PWM signal before breakage is shown in Figure 17.

Figure 18 proves that limiting the PWM signal did not stall the overheating of the wires; instead, overheating occurred earlier (342 s vs. 362 s). Analyzing the PWM signals, even though the maximum values were limited, the mean PWM values were greater (29.62%) than before (29.31%). Hence, greater PWM peaks seem to be a consequence more than a cause of SMA wire overheating.

The second approach consisted of trying to prevent overheating by including cooling times between actuation cycles. Different cooling times were evaluated (1 min, 1 min 30 s, 2 min and 2 min and 30 s) to check the lowest cooling time required for the wires to cool down. Only 30 s gaps were evaluated to ease the future usability of the device by health professionals; creating waiting times with decimals could be burdensome for future users and applications. Analyzing the breakage trends, it was observed that overheating always occurred after 5 min. Thus, it was decided to alternate between 5 min of actuation and different cooling times. After several experiments, it was observed that a gap of 1 min and 30 s was the lowest time that enabled actuator recovery. The results are shown over the 5 km/h pattern with a period of T = 12 s.

Figure 19 shows the second cooling down period of the wire. It can be observed how the actuator recovered over 1 min 30 s and how actuation was completely normal after this cooling time. Hence, it can be stated that allowing the actuators to cool for 1 min and 30 s enables their recovery so they can work for longer periods of time when tracking reference signals whose frequencies are unreachable when working in a continuous way.

Moreover, the same procedure was followed for actuators after reaching this overheating point. It was observed that if one SMA wire overheated, leaving it enough time to cool down enabled its activation for a similar number of working cycles before overheating occurs again. This cooling down gap was measured using the same procedure as before, obtaining a recovery time of 2 min between activations. Figure 20 shows the overheating of the actuator, its recovery and the new activation.

## 5. Discussion

Firstly, the use of SMA actuators as artificial muscles is analyzed and discussed in this section. It has been proven that SMAs are a feasible material for the proposed application, achieving a controlled and smooth behavior. Moreover, the specifications in terms of frequency and working time have been described for each walking pattern along with the mean error expected for each. From these specifications, professionals can decide between different working modes based on their technical needs. Additionally, it can be stated that walking patterns can be tracked via the ankle at acceptable frequencies with minimum errors in terms of angular positions, meeting the requirements of the proposed applications. However, there are different disadvantages that should be considered during the discussion. The first one concerns the delay experienced by the actuators when compared with the reference provided. This delay was measured, resulting in a mean value of 0.5 s and a square variability of ±0.03 s along the cycle with respect to other actuators. Hence, this delay is shared by all the structures, and thus compensated for during the whole walking cycle, displacing the performance by 0.5 s, but with no further severe consequences.

The second one relates to the frequencies achieved. It must be highlighted once again that the amplitude of the pattern was modulated depending on the walking velocity and not its frequency. Frequency was selected to be independent of the walking velocity for two main reasons. The first one addresses spasticity and other problems related to patients suffering from CP; these patients benefit from slow rehabilitation patterns. The second one is mobility; patients with a lack of strength and flexibility should achieve the desired ranges of motion before performing natural walking cycles in more realistic environments. However, high actuation frequencies are not feasible yet, although the on–off approach of the actuator allows it to perform better at higher frequencies. Future guidelines, including for a multi-wire actuator like the one developed in [45], are proposed for enhancing this working cycle.

Secondly, the design of the exosuit for the pediatric population, along with actuator fastening and routing, enables precise movements with a compact, flexible and easy-to-wear design. Some disadvantages were also observed in the design. The most important one concerns the suspended patient weight. Future guidelines should focus on the development of an exoskeleton for more advanced stages of therapy, where the patient can stand still, and the actuators must provide the accurate torque to each joint depending on the joint stiffness and the percentage of the patient’s weight supported by the device. Hence, the current results should be addressed and expanded to turn the low-level controller into an SMA-based impedance controller to improve the developed and validated soft actuator.

Finally, future guidelines should also include the adaptation of these results to an SMA-based device design for the knee flexo-extension and synchronize both performances to create a combined walking pattern.

## Figures and Tables

**Figure 1 biomimetics-09-00376-f001:**
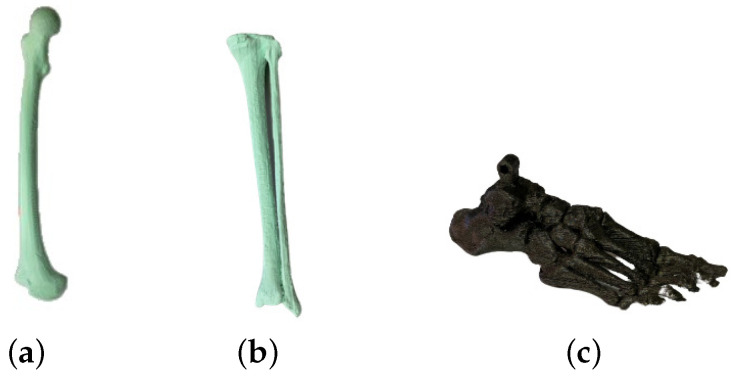
Bone segments printed in PLA. (**a**) Femur. (**b**) Tibia. (**c**) Foot.

**Figure 2 biomimetics-09-00376-f002:**
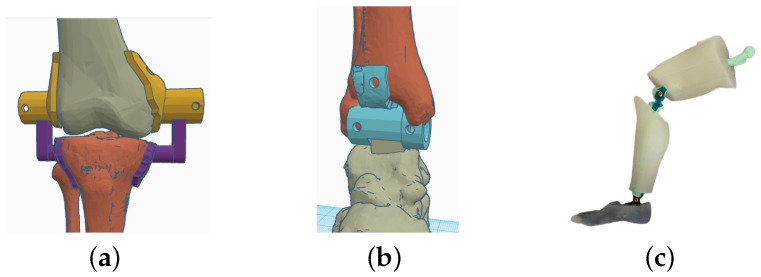
Leg joints. (**a**) Knee. (**b**) Ankle. (**c**) Whole Leg.

**Figure 3 biomimetics-09-00376-f003:**
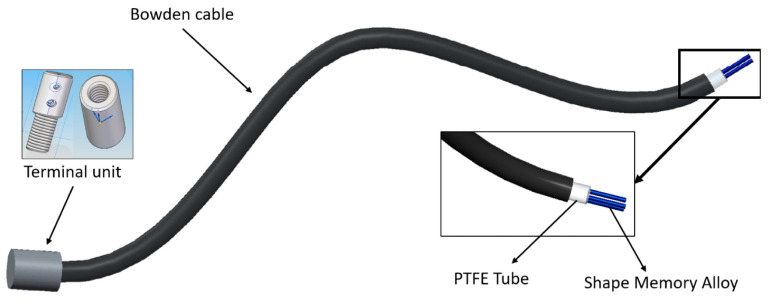
SMA actuator scheme.

**Figure 4 biomimetics-09-00376-f004:**
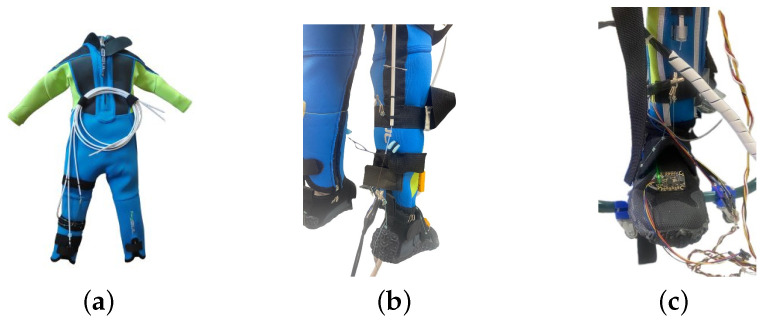
Exosuit. (**a**) SMA wires. (**b**) Ankle detail. (**c**) IMU detail.

**Figure 5 biomimetics-09-00376-f005:**
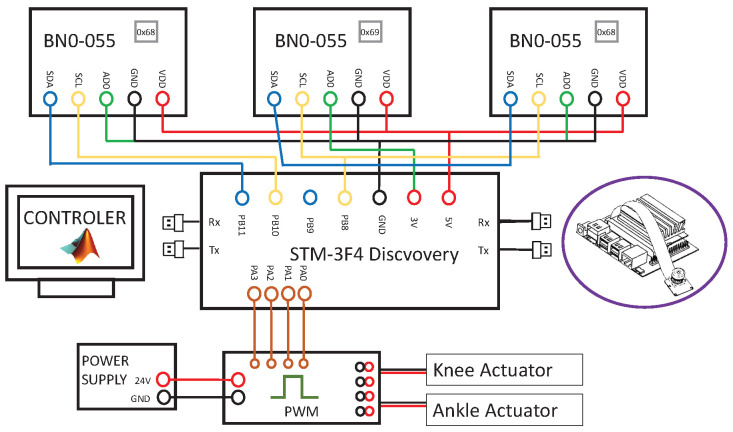
Hardware connections for one leg.

**Figure 6 biomimetics-09-00376-f006:**
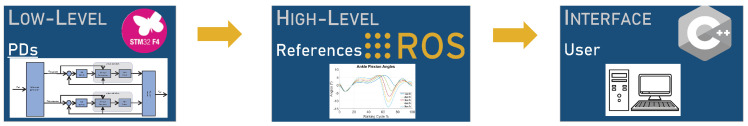
Two−level controller scheme.

**Figure 7 biomimetics-09-00376-f007:**
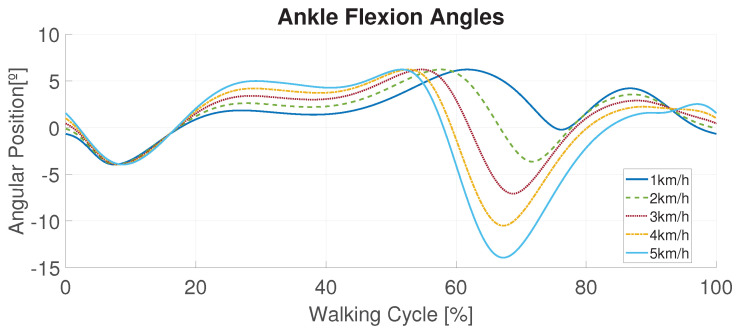
Reference right ankle for each walking pattern velocity. Positive: dorsiflexion. Negative: plantar flexion.

**Figure 8 biomimetics-09-00376-f008:**
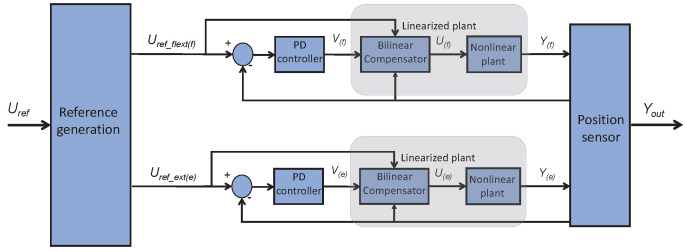
Agonist-antagonist control scheme with two B-PID controllers actuating in parallel.

**Figure 9 biomimetics-09-00376-f009:**
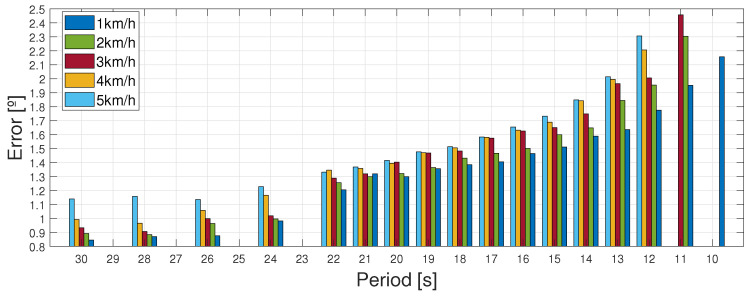
Mean error in absolute terms for each pattern and each period. Each column represents a different pattern, from 1 kmh^−1^ to 5 kmh^−1^.

**Figure 10 biomimetics-09-00376-f010:**
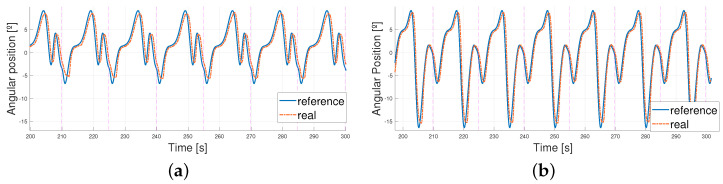
Real angular position of the ankle vs. reference T = 15 s. (**a**) 1 km/h pattern. (**b**) 5 km/h pattern.

**Figure 11 biomimetics-09-00376-f011:**
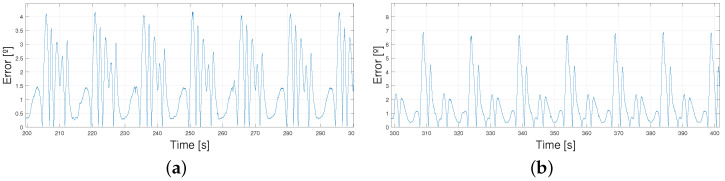
Absolute error between real angular position of the ankle and reference T = 15 s. (**a**) 1 km/h pattern. (**b**) 5 km/h pattern.

**Figure 12 biomimetics-09-00376-f012:**
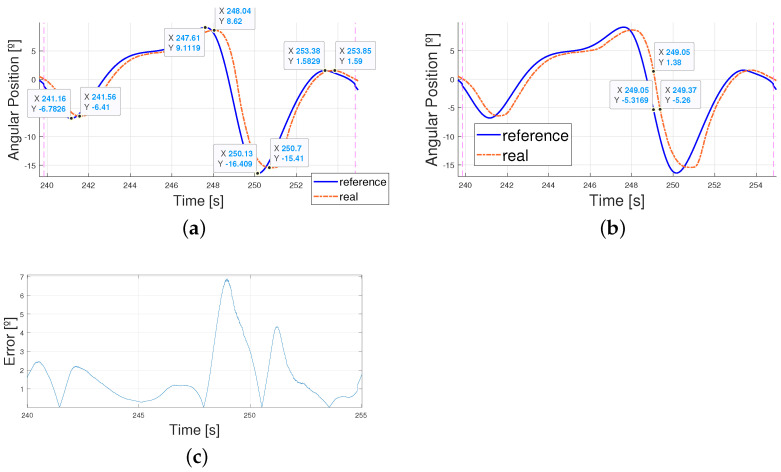
Zoomed in view of 240 s to 255 s (one period of T = 15 s). (**a**) Real angular position of the ankle vs. reference. Differences between peaks. (**b**) Real angular position of the ankle vs. reference. Phase lag. (**c**) Absolute error between real angular position of the ankle and reference.

**Figure 13 biomimetics-09-00376-f013:**
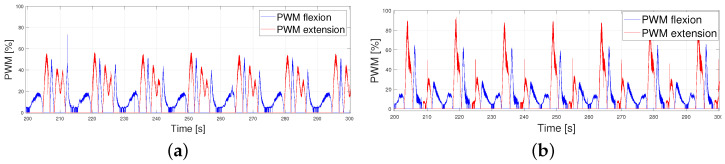
PWM signals for the activation of the flexor and extensor actuators, T = 15 s. (**a**) 1 km/h pattern. (**b**) 5 km/h pattern.

**Figure 14 biomimetics-09-00376-f014:**
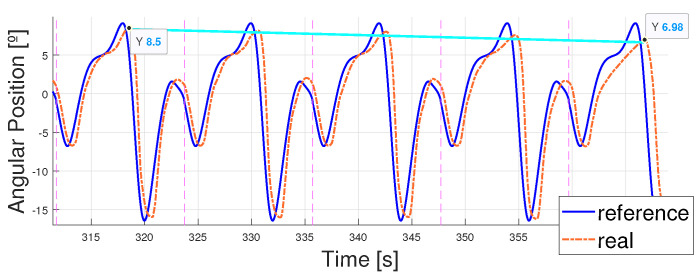
Absolute error between real angular positions of the ankle and reference. Overheating of the SMA wire actuator. T = 12 s.

**Figure 15 biomimetics-09-00376-f015:**
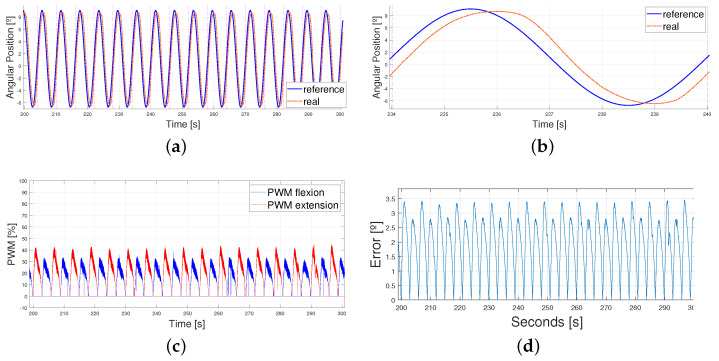
Results obtained from real data when following a sinusoidal reference of T = 6 s. Results are shown from 200 s to 300 s. (**a**) Sinusoidal reference vs. real angular position of the ankle. (**b**) Sinusoidal reference vs. real angular position of the ankle. Zoomed in to (234–240 s) (**c**) PWM signals for flexion and extension controllers. (**d**) Absolute error between the sinusoidal reference and the real angular position of the ankle.

**Figure 16 biomimetics-09-00376-f016:**
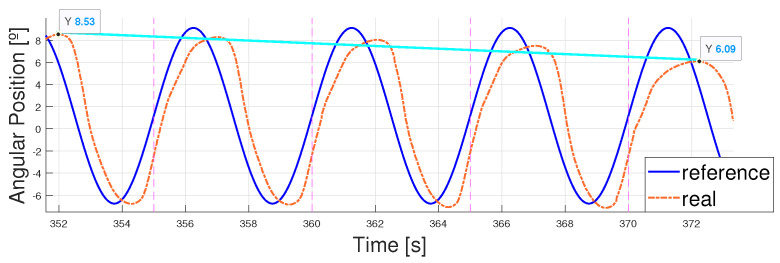
Sinusoidal reference vs. real position of the ankle with SMA wire break down. T = 5 s. Breakage occurs at 372 s.

**Figure 17 biomimetics-09-00376-f017:**
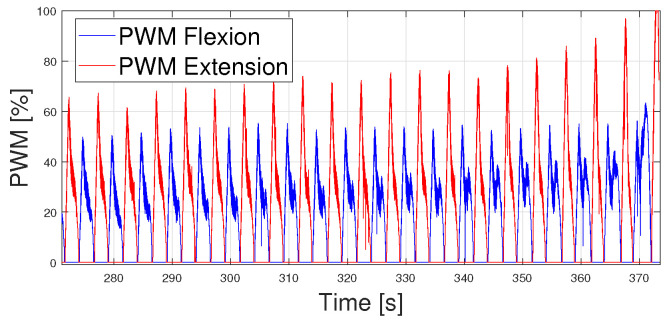
PWM signals for flexion and extension controllers tracking a sinusoidal reference. T = 5 s.

**Figure 18 biomimetics-09-00376-f018:**
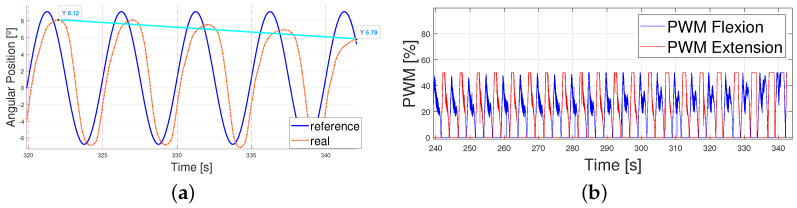
Results obtained from real data when following a sinusoidal reference of T = 6 s. Results are shown from 200 s to 300 s. (**a**) Sinusoidal reference vs. real angular position of the ankle. Breakage occurs at 342 s. (**b**) Limited PWM signals for flexion and extension controllers.

**Figure 19 biomimetics-09-00376-f019:**
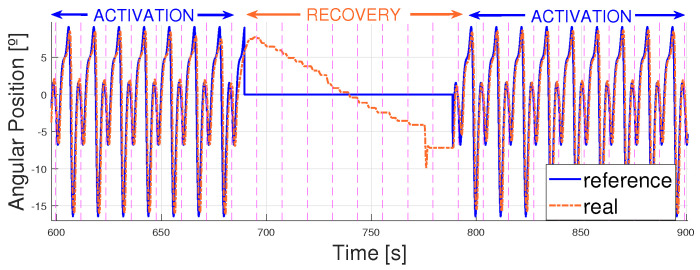
5 km/h pattern reference vs. real angular position of the ankle. T = 12 s. Signal between 600 and 900 s. Activation time (5 min) vs. recovery time (1 min and 30 s).

**Figure 20 biomimetics-09-00376-f020:**
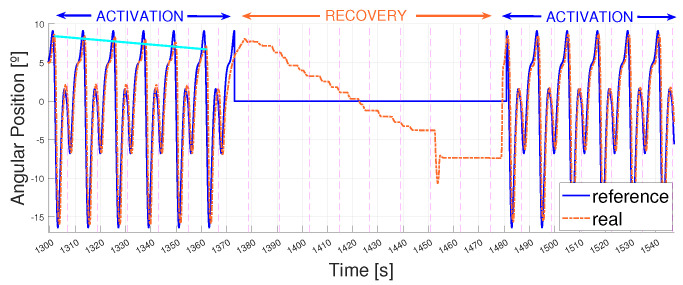
5 km/h pattern reference vs. real angular position of the ankle. T = 12 s. Signal between 1300 and 1550 s. Activation time (until overheating) vs. recovery time (2 min).

**Table 1 biomimetics-09-00376-t001:** Comparative body measurements for a 3-year-old child. Real vs. dummy.

Segment	Length [cm]		Weight [kg]		Diameter [cm]	
	**Real**	**Dummy**	**Real**	**Dummy**	**Real**	**Dummy**
Thigh	22.5	22.8	1.55	1.43	7.5	7.3
Shank	18.8	17.8	0.65	0.61	5.0	5.1
Foot	14.4	14.2	0.23	0.25	3.0	3.0

**Table 2 biomimetics-09-00376-t002:** Functional angles during the walking cycle.

Movement	Knee	Ankle
Flexion (^∘^)	60	5
Extension (^∘^)	0	15

**Table 3 biomimetics-09-00376-t003:** Technical specifications of the selected SMA wire.

Joint	Ankle
Diameter (mm)	0.38
Activation Temperature (^∘^C)	90
Payload (kg)	2.25
Current (A)	2.25
Resistance (Ω/m)	8.3

**Table 4 biomimetics-09-00376-t004:** Mean error in absolute terms for each pattern and each period. Bold numbers represent the periods at which the actuators overheated and broke down.

Period [s]	Error [^∘^]				
9	-	-	-	-	-
10	**2.156**	-	-	-	-
11	1.953	**2.305**	**2.457**	-	-
12	1.776	1.954	2.006	**2.207**	**2.306**
13	1.636	1.845	1.966	1.996	2.014
14	1.589	1.648	1.749	1.843	1.848
15	1.512	1.599	1.651	1.689	1.733
16	1.464	1.502	1.625	1.633	1.665
17	1.404	1.466	1.575	1.580	1.583
18	1.386	1.432	1.482	1.506	1.513
19	1.356	1.365	1.468	1.470	1.478
20	1.299	1.321	1.404	1.396	1.415
21	1.320	1.299	1.320	1.359	1.369
22	1.205	1.257	1.289	1.345	1.332
24	0.981	0.996	1.021	1.166	1.227
26	0.875	0.963	0.999	1.057	1.136
28	0.870	0.884	0.907	0.965	1.159
30	0.845	0.892	0.932	0.992	1.143
**Pattern**	1 km/h	2 km/h	3 km/h	4 km/h	5 km/h

**Table 5 biomimetics-09-00376-t005:** Absolute error for a sinusoidal sign at different periods. Bold numbers represent the periods at which the actuators overheated and broke down.

Period [s]	Error [^∘^]	Period [s]	Error [^∘^]	Period [s]	Error [^∘^]
4	-	10	1.312	17	0.795
5	**2.499**	11	1.217	20	0.989
6	2.094	12	1.149	24	0.863
7	1.759	13	1.022	26	0.896
8	1.589	14	0.983	28	0.757
9	1.414	15	0.832	30	0.765

## Data Availability

The original contributions presented in the study are included in the article, further inquiries can be directed to the corresponding author.

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
