# Peer review of "Design of an SMA-Based Actuator for Replicating Normal Gait Patterns in Pediatric Patients with Cerebral Palsy"

_biomimetics, 2024, doi:10.3390/biomimetics9070376_

Round 1

Reviewer 1 Report

Comments and Suggestions for Authors

This paper presents the design details and preliminary experimentation on a Shape Memory Alloy-based actuator for patients having cerebral palsy. 

- Please reduce the background details from the Abstract. Instead, elaborate more on the methodology and more importantly, summarise in 1-2 sentences on the results and performance achieved in a quantitative manner.

- Please make the representation consistent in the paper. e.g. paper outlines mention I, II, III while the actual numbering of the Sections list 1, 2, 3 ...

- Mention the version of MATLAB/Simulink used in this research. Also, list the specifications of the machine (Computer) used to run the simulations.

- Limitations of existing exoskeleton systems mentioned on LIne 44-45 can benefit from a reference 10.1016/j.bbe.2014.01.003.

- Figures containing the plots/graphs need to be redrawn using distinguishing markers. So, e.g., plot quantity 1 with a solid line, quantity 2 with a dotted line, quantity 3 with a dashed line, quantity 4 with a dotted-dashed line and so on. Please do not refer to the waveforms based on their color.

- Text size in some of the figures is not readable e.g. Pease see Figure 8.

- 's' or 'Seconds'? On x-axis of graphs, please use Time [s] instead of Seconds [s].

- Arrange the list of Abbreviations on Page 16 in alphabetical order.  

Comments on the Quality of English Language

Moderate editing of English language required.

Reviewer 2 Report

Comments and Suggestions for Authors

Dear Authors,

thank you for submitting the manuscript titled: Design of a SMA-based Actuator for Replicating Normal Gait Patterns in Pediatric Patients with Cerebral Palsy for the review process. The article seems interesting at first sight, but it should be corrected according to the below suggestions:

- Authors used IMU to obtain the real angular position of both angles. Please write a few sentences in the Introduction about IMUs. For example in the article titled:  Gait Recognition: A challenging Task for MEMS Signal Identification, Smart Innovation, Systems and Technologies, 2019, KES-SDM, 473-84, authors used Inertia System technology (consisting of 6 IMUs) to obtain kinematic parameters of human gait. To improve article quality I suggest citing the above manuscript and writing one paragraph about IMUs.

-  Table 2. I think it will be better not to include the max angles. The authors present actuators for patients with CP, thus the max angles will not be obtained.

- Please connect Table 1 and Table 3 (Column real and dummy)

- Please include parameters of IMU sensors (e.g sampling)

- Lack of limitation of the study
